# GLSO: Grammar-guided Latent Space Optimization for Sample-efficient Robot Design Automation

**Jiaheng Hu**
Robotics Institute
Carnegie Mellon University
jiahengh@andrew.cmu.edu

**Julian Whitman**
Department of Mechanical Engineering
Carnegie Mellon University
jwhitman@andrew.cmu.edu

**Howie Choset**
Robotics Institute
Carnegie Mellon University
choset@andrew.cmu.edu

**Abstract:** Robots have been used in all sorts of automation, and yet the design of robots remains mainly a manual task. We seek to provide design tools to automate the design of robots themselves. An important challenge in robot design automation is the large and complex design search space which grows exponentially with the number of components, making optimization difficult and sample inefficient. In this work, we present Grammar-guided Latent Space Optimization (GLSO), a framework that transforms design automation into a low-dimensional continuous optimization problem by training a graph variational autoencoder (VAE) to learn a mapping between the graph-structured design space and a continuous latent space. This transformation allows optimization to be conducted in a continuous latent space, where sample efficiency can be significantly boosted by applying algorithms such as Bayesian Optimization. GLSO guides training of the VAE using graph grammar rules and robot world space features, such that the learned latent space focus on valid robots and is easier for the optimization algorithm to explore. Importantly, the trained VAE can be reused to search for designs specialized to multiple different tasks without retraining. We evaluate GLSO by designing robots for a set of locomotion tasks in simulation, and demonstrate that our method outperforms related state-of-the-art robot design automation methods.

**Keywords:** Robot Design Automation, Latent Optimization, Graph Grammar

## 1 Introduction

Robot design automation aims to discover robot body structures that optimize a given objective. While this subject has been long-studied [1, 2, 3, 4, 5], the problem is difficult due to the large search space and the computational expense involved in evaluating candidate designs. Classic design automation approaches resort to discrete black-box optimization techniques such as Genetic Algorithms (GA)[1], Genetic Programming (GP) [3], and Random Graph Search (RGS)[4]. While these methods typically work well when the objective function is inexpensive to evaluate (e.g., obtaining the locomotion speed of a simple robot in simulation with a pre-defined controller), they require a large number of objective function evaluations and are not suitable for situations where evaluation is expensive (e.g., through real-world evaluation, or creating customized dynamic motion plans for each new design). Recent works utilize graph grammar rules to confine the search space [5, 1], such that the number of evaluations can be reduced. However, they still operate in a high-dimensional combinatorial search space and require a considerable number of sample evaluations.

In this work, we introduce Grammar-guided Latent Space Optimization (GLSO), a framework for sample-efficient robot design automation. Given a robot design space defined by the possible combinations of a set of discrete primitive components, GLSO first learns a low-dimensional, continuous

6th Conference on Robot Learning (CoRL 2022), Auckland, New Zealand.

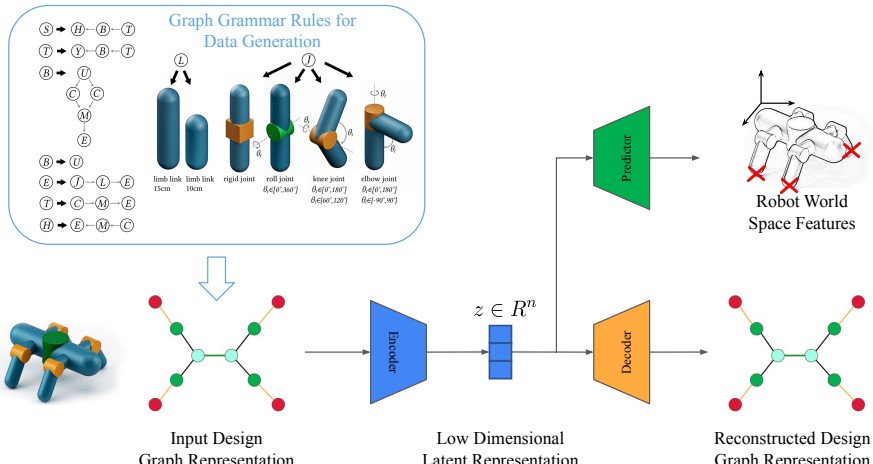

Figure 1: Our framework begins by collecting a dataset of robot designs based on a set of graph grammar rules, as shown on the top left. This process of enumerating designs is computationally inexpensive, as no controller is needed. The collected data (example on bottom left) is subsequently used to train a graph variational autoencoder (VAE), which defines a mapping between a low dimensional continuous latent space and the combinatorial design space. A property predictor (shown as the green trapezoid) is simultaneously trained to predict the world space features grounding of the robots from the latent vector, in order to encourage physically similar robots to be grouped together in the latent space. After the VAE is trained, optimization can be performed in the latent space in search of high-performing designs. This VAE can further be used for multiple distinct tasks without the need for retraining.

representation of the robot design space through unsupervised learning. The learned representation allows us to then convert the combinatorial design automation search into a continuous optimization problem, where we apply sample-efficient Bayesian Optimization (BO) [6] to search in the latent space for high-performing designs. GLSO uses a graph variational auto-encoder (VAE) to learn mappings between the design space and latent space. Importantly, the learned latent space can be used to optimize designs for multiple different tasks without the need for retraining.

GLSO is inspired by recent advancements in molecule synthesis [7, 8]. However, unlike molecule optimization, where existing large-scale datasets such as ZINC [9] are readily available to supervise training, the domain of robot design has no such dataset available. Instead, GLSO generates training data by leveraging graph grammar rules [5], which confine the search space and implicitly inject a prior into our learned latent representation.

An additional challenge associated with robot design automation is the potentially "chaotic" objective function, i.e. that two structurally similar robots may have very different performance for a given task. For example, consider the designs shown in Fig. 3b: they have nearly the same design graph, but they all have drastically different locomotive performance. We address this problem by including an additional objective in our VAE training, which ensures that neighboring designs in the latent space not only have similar graphs, but also share similar world space features, such as the bounding box of the robot or the end-effector workspace. That is, in addition to the conventional VAE encoder/decoder reconstruction loss, GLSO simultaneously trains a neural network to learn world space feature grounding the robot designs, which we called the property prediction network. The complete pipeline can be seen in Fig. 1[1].

We evaluate GLSO by designing robots for a set of locomotion tasks, with a component library that includes different joints and links with various rotational angles and axes, sizes, and weights. Our method outperforms state-of-the-art robot design automation methods, consistently identifying higher-performing designs when given the same number of sample evaluations[2].

---

[1]Upper left portion of the figure adopted from Xu et al. [10] with author's approval

[2]The code used in this work is available at `https://github.com/JiahengHu/GLSO`.

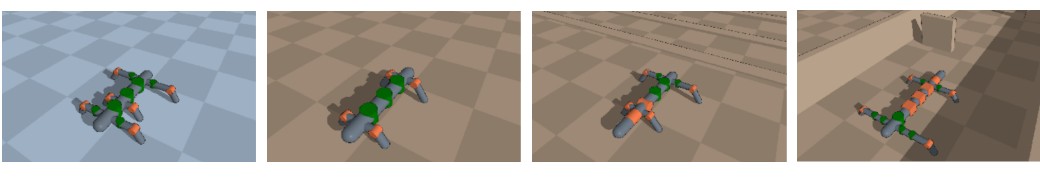

| (a) Frozen Lake Task | (b) Flat Terrain Task | (c) Ridged Terrain Task | (d) Wall Terrain Task |

Figure 2: We evaluate our framework on a set of locomotion tasks as shown in the figures. The above images showcase the best design identified by our method for four tasks. A video showing the motion of these robots is provided in the supplement.

## 2   Related Work

### 2.1   Evolutionary Algorithms

A *de facto* choice for robot design automation is population-based stochastic optimization algorithms, such as evolutionary algorithms (EAs), due to their ability to search combinatorial design spaces [1, 2, 4, 11, 12, 13, 14, 15]. Specifically, EAs maintain and update a population of candidate solutions through a set of operations (e.g., mutations, cross-over, evaluation, and elite selection) to search for high-quality designs. However, EAs require a large number of objective function evaluations, and can quickly become computationally infeasible when the objective function evaluation is costly, such as when the determining the optimal controller can be computationally burdensome, or when the evaluation needs to take place in real world instead of in simulation.

### 2.2   End-to-end Design Generation

Another class of design automation methods uses learning-based approaches to train a design generator (typically in the form of a neural network) to predict suitable designs for a given task [16, 17, 18]. Learning-based approaches are particularly promising for field deployment due to their near real-time execution speed. However, they require computationally expensive training procedures in which many different combinations of tasks and designs must be tested. During this training, they apply strong assumptions about the set of tasks for which the robot will be specialized. Furthermore, recent works also assume a restrictive design space (e.g., fixed-topology graph, binary 3D voxel grid), and it is unclear how end-to-end approaches can be extended to more complex designs.

### 2.3   RoboGrammar

Graph grammar rules provide an effective means to restrict a combinatorial design space, which can improve optimization efficiency, especially when the design space is high-dimensional [5, 1]. In robotics, graph grammars have been applied to model both physical structure as well as control laws [19, 20, 21]. More recently, RoboGrammar [5, 10] introduced a set of recursive graph grammar rules for robot design automation. RoboGrammar operates on graphs composed of terminal and non-terminal symbols, where non-terminal symbols are temporary nodes and terminal symbols correspond to physical robot components. Starting from a nonterminal symbol "S", RoboGrammar iteratively applies available rules until reaching a acyclic graph of terminal symbols. The final graph corresponds to a robot design, where the hardware components are represented as nodes and the physical links are represented as edges. The upper left corner of Fig. 1 demonstrates the set of rules proposed by RoboGrammar. To optimize the robot designs based on the grammar rules, RoboGrammar proposed Graph Heuristic Search (GHS), which operates on a search tree defined by the graph grammar rules. The search is conducted in an A* like manner, where a heuristic function in the form of a graph neural network [22] is trained during the search process. For more details about the RoboGrammar method, see Zhao et al. [5].

### 2.4   Latent Space Optimization

Latent space optimization (LSO) is a framework designed to extend continuous optimization techniques to combinatorial search spaces [23, 24]. LSO first learns a generative model (typically a

variational autoencoder) [25] $g : \mathcal{Z} \rightarrow \mathcal{X}$ which maps data points from a continuous latent space $\mathcal{Z}$ to the combinatorial search space $\mathcal{X}$. Continuous optimization can subsequently be performed in the latent space $\mathcal{Z}$ using standard continuous optimization algorithms such as BO. LSO has been shown to be an effective framework for domains involving discrete data, including natural language [26], arithmetic expressions [7], programs generation [27] and molecules synthesis [28, 8]. In the domain of robotics, Spielberg et al. [29, 30] facilitated design and control co-optimization for soft robots by learning a low-dimensional latent representation. For graph-based representation of robot designs, Kim et al. [31] made initial progress in learning a latent representation of robot morphology, but only with a small set of serial-chain topology designs, and did not perform design optimization within that latent space.

## 2.5 Graph Neural Networks for Robotics

Graph Neural Networks (GNN) [22] are a neural network architecture which operates on graph-structured inputs. For robots with graph-based representation, GNNs have been used to control different designs by abstracting the robots as graphs [32, 33, 34, 35]. In our work, we utilize GNNs as a premutation-invariant process to learn a compressed continuous representation of graph-structured robot designs.

## 3 GLSO: Learning Latent Design Representation

GSLO trains a generative model that defines a mapping between a continuous latent space and the discrete design space. Similar to previous LSO works, our method uses a VAE as a generative model. The VAE consists of an encoder (section 3.2) and a decoder (section 3.3). In addition to the encoder and decoder, we co-train a property prediction network (section 3.4) to predict world space features of the robots. The goal of co-training the property predictor is to encourage designs with similar physical properties to be close together in the latent space; we find that in turn, this makes downstream design optimization in the latent space more efficient. The training data for the VAE is generated through recursively applying a set of graph grammar rules (section 3.1). The grammar ensures that the training designs are valid, which implicitly bias the mapping from latent space to design space towards promising robots. Importantly, the trained VAE can be reused to search for designs specialized to multiple different tasks without retraining.

**Notation.** Each design is represented as an acyclic graph $\mathcal{G} = (V, E)$ where $V$ corresponds to the hardware components (nodes) and $E$ the connectivity between them (edges). Note that an acyclic design graph can also be viewed as a tree, where the root node corresponds to the one component on the body, and the each of the leaf nodes correspond to the robot's end-effectors. We use $i, j, k$ to refer to nodes indices, and define $N(i)$ as the neighbor of a node $i$. We denote the sigmoid function as $\sigma(\cdot)$ and the ReLU function as $\tau(\cdot)$, and trainable weights in the models as $W$ and $U$.

## 3.1 Data Collection via Graph Grammar

For a given set of robot hardware components, GSLO begins by collecting a dataset of robots composed of these hardware components. This dataset will then be used to train the graph VAE. We desire a dataset that covers much of the design space, while containing few invalid designs.

One key idea behind GSLO is that graph grammar rules can serve as a guiding tool to generate a large dataset of designs, due to their ability to filter out invalid designs and to produce a tractable and meaningful subspace of designs. Specifically, we adopt the set of grammar rules proposed in RoboGrammar [5]. To generate a random design using RoboGrammar, we start from the start symbol "S" and randomly apply valid grammar rules until the design graph consists of only terminal nodes. Each terminal node corresponds to a specific hardware component, and therefore each design graph has a one-to-one correspondence with a physical robot. We additionally collect each design's contact locations when that design is placed at rest on flat ground in simulation. These contact locations will be used to train the property predictor in section 3.4. Since the data collection process does not involve creating controllers, it can be readily applied to a new set of grammar rules / hardware components. In our experiments, the data collection process produces around $5e^5$ designs in approximately four hours, and could be parallelized for additional speedup.

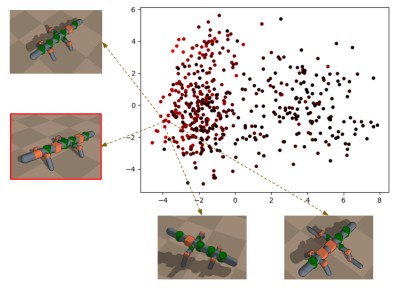

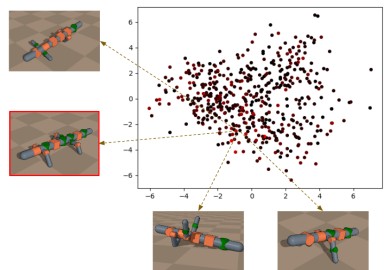

(a) Latent Space with PPN    (b) Latent Space without PPN

Figure 3: Visualization of two different latent spaces (a) when the VAE is co-trained with the property predictor, and (b) without the property predictor. Here we project the latent points into a two-dimensional space, for visualization purposes, by taking the first two principal components. The color of the points correspond to its performance for a given task (here, traversing flat terrain), where a darker color denotes worse performance. We additionally show designs neighboring the same robot (with red border) in the two different latent spaces. We can see that the addition of the property predictor implicitly encourages designs with similar performance to be grouped together, which we find results in more efficient optimization.

## 3.2    Robot Encoder

Since the robots are represented as graphs, we use a graph autoencoder to encode the robots via a graph message passing neural network (MPNN)[36, 8]. MPNNs operate on graph-structured inputs, where hidden representations of the graph nodes are updated iteratively via messages sent along graph edges [22]. Each node $v_i$ contains a one-hot feature vector $x_i$ indicating its type. We denote a message from a node $v_i$ to $v_j$ as $m_{i,j}$, and $m_{j,i}$ vice versa.

At each message passing iteration, messages are updated as:

$$m_{i,j} = \text{GRU}(x_i, \{m_{ki}\}_{k \in N(i)/j}) \tag{1}$$

where GRU refers to Gated Recurrent Unit [37] adapted for graph message passing. For detailed formula of the GRU update, please refer to the supplementary material.

After $t$ iterations of message passing, we obtain the latent representation of each node by aggregating its inward messages,

$$h_i = \tau(W^e x_i + \sum_{k \in N(i)} U^e m_{ki}). \tag{2}$$

The robot graph representation is calculated as the sum of the latent representation of the leaf nodes,

$$h_{\mathcal{G}} = \sum h_{\text{leaf}}. \tag{3}$$

Finally, the mean $\mu_{\mathcal{G}}$ and variance $\sigma_{\mathcal{G}}$ of the variational posterior approximation are computed from $h_{\mathcal{G}}$ by applying two separate affine layers, where the latent vector $z_{\mathcal{G}}$ is sampled from a Gaussian distribution $z_{\mathcal{G}} \sim \mathcal{N}(\mu_{\mathcal{G}}, \sigma_{\mathcal{G}})$

## 3.3    Robot Decoder

The robot decoder maps the continuous latent vector $z_{\mathcal{G}}$ back to a robot design tree $\mathcal{G}$ by sequentially adding nodes to a partial design tree in depth-first order, starting from the root. For every visited node, the decoder first predicts whether it has children to be generated. If so, a new node is created and attached to the current node, with its node type predicted by the decoder. The messages of existing nodes will not be cleared during decoding. This procedure is recursively applied to the newly created nodes until the current node has no more children to generate, where the decoder backtracks to its parent node and repeat this process.

The decoder makes predictions based on message propagation in the current partial tree at each time step. The messages are propagated using the same GRU structure as in the encoder. The decision

of whether to create new node at each time step is predicted as a probability $p_t$ based on the inward messages $h_{k,i}$, latent vector $z_{\mathcal{G}}$ and the node features $x_i$,

$$p_t = \sigma(u^c \cdot \tau(W_1^c x_{i_t} + W_2^c z_{\mathcal{G}} + W_3^c \sum_k h_{k,i_t})). \tag{4}$$

For a new node $j$ created from parent $i$, the label $q_j$ is predicted with,

$$q_j = \text{softmax}(U^l \tau(W_1^l z_{\mathcal{G}} + W_2^l h_{i,j})). \tag{5}$$

The decoder is trained to maximize the reconstruction likelihood $p(\mathcal{G}|z_{\mathcal{G}})$. Let $\hat{p}_t$ and $\hat{q}_j$ be the ground truth values of $p_t$ and $q_j$ respectively, the decoder loss is:

$$\mathcal{L}_d(\mathcal{G}) = \sum_t \mathcal{L}_c(p_t, \hat{p}_t) + \sum_j \mathcal{L}_l(q_j, \hat{q}_j), \tag{6}$$

where $\mathcal{L}_c$ and $\mathcal{L}_l$ are cross-entropy losses. Similar to language generation, we apply *teacher forcing* [38] during training, where ground truth topology and labels are used at each step for prediction.

### 3.4 Property Predictor

We hypothesize that a learned latent space in which designs with similar capabilities are close together, and which contains few invalid designs, will allow for more efficient optimization. To obtain such a latent space, we co-train a property prediction network (PPN) with the VAE. The PPN maps the mean of the variational encoding distribution $\mu_{\mathcal{G}}$ to the corresponding robot's world space features, which implicitly encourages robots with similar world space features to have similar latent representations. In this work, since we are primarily experimenting with locomotion tasks, we define the world space features as a vector $v_{\text{contact}}$ consisting of the robot's 2D contact locations on flat ground. The contact locations are sorted based on their x values and zero padded to a fixed length to form $v_{\text{contact}}$. $v_{\text{contact}}$ is collected together with the robot data as described in section 3.1. The PPN is trained to minimize the mean square error between the predicted contact vector $\hat{v}_{\text{contact}}$ and the ground truth contact vector $v_{\text{contact}}$, i.e. $\mathcal{L}_{PPN} = ||\hat{v}_{\text{contact}} - v_{\text{contact}}||^2$.

**Training:**

The final loss of the graph VAE is a weighted sum of the decoder loss $\mathcal{L}_d$, the PPN loss $\mathcal{L}_{PPN}$, and the KL-divergence $\mathcal{L}_{KL}$ between the variational posterior and the prior latent distribution $\mathcal{N}(0, I)$:

$$\mathcal{L}_{vae} = \mathcal{L}_d + \mathcal{L}_{KL} + \lambda \mathcal{L}_{PPN} \tag{7}$$

We minimize $\mathcal{L}_{vae}$ using gradient descent. Training took around five hours on a NVIDIA GTX 1070 graphics card. We provide the detailed training hyperparameters, as well as visualization of interpolation in the learned latent space, in the supplementary materials.

## 4 GLSO: Optimization in the Learned Latent Space

The trained VAE associates each robot design with a latent vector, given by the mean of the variational encoding distribution $\mu_{\mathcal{G}}$. We can use the VAE to transform robot design automation from a combinatorial optimization problem into a continuous one. We then apply Bayesian Optimization to search for the latent vector where the associated robot design has the highest performance.

### 4.1 Controller & Evaluation

To evaluate the performance of a given robot structure, we need to derive its controls. This reveals one challenge in robot design optimization– hidden inside each evaluation of the design optimization objective function is a trajectory or policy optimization problem, which itself is often computationally expensive. In this work, we utilized model predictive control (MPC) [39] to create controllers for different robot structures across different terrains. MPC predicts future behaviour using a model of the system, optimizes controls for a finite horizon, executes a small number of the optimized control, and then replans. Specifically, we used model predictive path integral control (MPPI) [40], a sampling based MPC method for controlling the generated robot designs. Our implementation follows [5]. Each robot is controlled in a simulation implemented using the Bullet Physics library [41]. We adopt the same objective function as in [5], where robots are rewarded for forward progression while maintaining its initial orientation. Computing the trajectory of each robot takes 30 - 60 sec, and is the primary computational bottleneck of the design optimization.

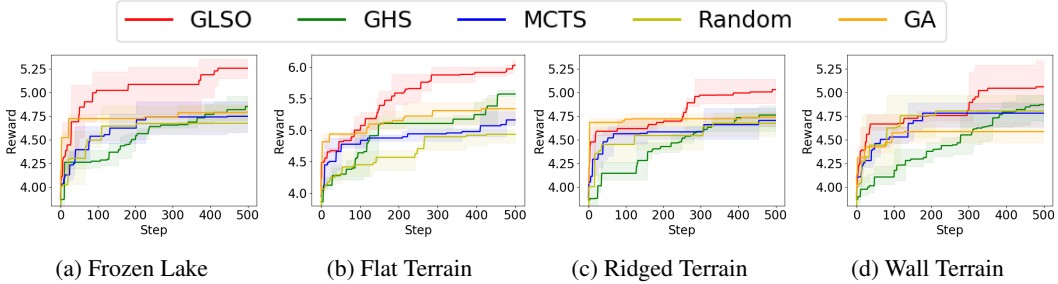

(a) Frozen Lake      (b) Flat Terrain      (c) Ridged Terrain      (d) Wall Terrain

Figure 4: Comparison against related methods. The solid line shows average over 3 different random seeds, and the error band represents the maximum and minimum. In each task, GLSO produces higher-reward designs within 500 steps (design samples).

## 4.2 Bayesian Optimization in Latent Space

We use Bayesian Optimization (BO) [6], a sample-efficient black-box optimizer, to perform optimization in the latent space. Specifically, our BO implementation uses Gaussian Processes (GP) [42] to build a model of the objective function, which includes both the current estimation of the function and the uncertainty around the estimation. We then use expected improvement (EI) as the acquisition function to determine the optimal point to sample next. The sampled point is subsequently evaluated and used to update the GP model. This procedure is repeated until we reach the computation limit, defined as the maximum number of objective function evaluations. We additionally apply domain reduction [43], where the bounds of the latent space are contracted during the optimization to reduce oscillation. The hyperparameters of the BO are provided in the supplementary material.

# 5 Experimental Results

We evaluate our method by generating designs for the following four locomotion tasks, each defined by its corresponding terrain:

- **Flat Terrain**: A flat plane with a friction coefficient of 0.9.
- **Frozen Lake Terrain**: A flat plane with a low friction coefficient of 0.05.
- **Ridged Terrain**: Includes hurdles that the robots must jump or crawl over.
- **Wall Terrain**: Includes high walls placed in a slalom-like manner.

The reward of each robot is measured as described in section 4.1. We report comparisons to previous approaches as well as ablated versions of our approach. For each each task, we allow each method a maximum of 500 objective function evaluations, i.e., control and trajectory optimization for up to 500 designs. Images of the tasks and optimized designs are shown in Fig. 2. Videos of their motions are in the supplementary material.

## 5.1 Comparisons and Baselines

To demonstrate the effectiveness of GLSO, we benchmark our method against the following:

**Random Search**: Random designs are generated using the graph grammar rules.

**Monte Carlo Tree Search (MCTS)**: A baseline adopted from [5], performing Monte Carlo Tree Search (MCTS) [44] on a search tree defined by the graph grammar rules.

**Graph Heuristic Search (GHS)**: A design automation method proposed in [5]. GHS performs search on the same graph grammar search tree as does MCTS. Our implementation follows [5].

**Genetic Algorithm (GA)**: Our implementation of Genetic Algorithm (GA) follows [4], where graph mutation with uncertainty is used to mutate the population at each iteration. We used a population size of 50 and evolved them until the number of evaluation limit was reached. Note that unlike the other comparison methods, GA does not operate in the grammar space, as crossover and mutation operations are not clearly defined for the graph grammar proposed by Zhao et al. [5].

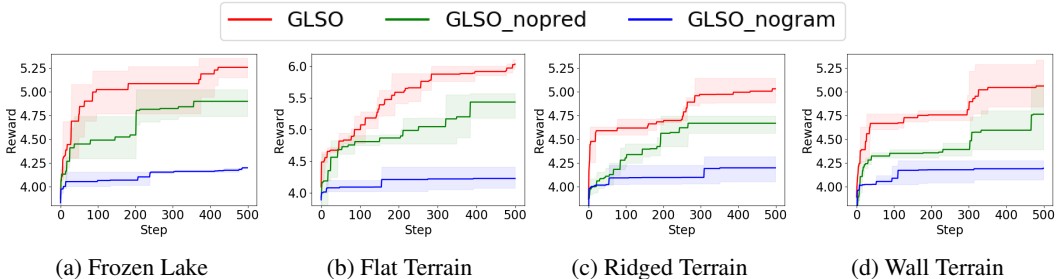

(a) Frozen Lake      (b) Flat Terrain      (c) Ridged Terrain      (d) Wall Terrain

Figure 5: Comparison of GLSO with ablated variants. The solid line shows average over 3 different random seeds, and the error band represents the maximum and minimum. These results show that both the graph grammar and the property predictor are important to GLSO.

The optimization curves for each of the tested algorithms is shown in Fig. 4. We found our method outperform all comparison methods and baselines.

## 5.2 Ablation Studies.

We performed ablation studies to investigate the effects of the graph grammar rules and the property prediction network. Results are shown in Fig. 5. In the "GLSO_nogram" test, we removed the graph grammar rules during data collection, and the VAE training set is created through random generation of topology and node labels. In the "GLSO_nopred" test, we removed the property prediction network during VAE training. We found that both the graph grammar and property predictor are crucial to the success of GLSO. Additionally, we created a visualization of how the property prediction network influences the latent space. Fig. 3 presents points in the latent space created by training the VAE with and without the property prediction.

## 6 Limitations & Conclusions

In this work, we proposed GLSO: a sample-efficient approach for optimizing robot designs based on latent encoding. We believe that our approach is general to the domain of robot design automation, and represents a step towards achieving efficient robot design automation, in particular when evaluation of each design is computationally expensive.

An important limitation of our approach is the requirement for a pre-defined set of graph grammar rules as well as world space features. Extending this work to different sets of hardware components and tasks will likely take some expert knowledge or domain intuition. Secondly, while we ultimately aim to extend design automation to real-world tasks, the designs in our current experiments do not correspond to existing physical hardware. An important next step will be to demonstrate GLSO on a set of modular robotic hardware components. Furthermore, for the evaluation steps to take place in real life, it will be necessary to further improve the sample efficiency of our methods, as 500 sample evaluations will likely prove to be too many designs to prototype with real-world hardware.

Besides the limitations above, there are also a few potential extensions of this work. Firstly, the learned latent representation of the designs may have other uses besides design automation. For example, down-stream learning tasks that require outputting a robot design [18, 16] may benefit from learning to output a continuous latent vector instead of a discrete graph. Secondly, recent advancement on latent space optimization, such as leveraging decoder uncertainty [45] and weighted retraining [23], may be beneficial to our optimization scheme. A future direction would be to explore how these techniques can be adapted to improve the performance of GLSO.

**Acknowledgments**

We thank the anonymous reviewers for their helpful comments on improving the paper. We thank the member of Biorobotics lab for their valuable feedback on idea formulation and manuscript.

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
