# OpenReview forum: "GLSO: Grammar-guided Latent Space Optimization for Sample-efficient Robot Design Automation"
_robot-learning.org/CoRL/2022/Conference — CoRL 2022 Poster_

### Official Review · Reviewer_t5Qe · 2022-07-26

**Originality:** Very Good
**Technical Quality:** Very Good
**Clarity Of Presentation:** Excellent
**Impact:** 4

**Recommendation:**

Strong Accept: I recommend accepting the paper and will argue for my recommendation even if other reviewers hold a different opinion.

**Summary:**

Design optimization of kinematic robots (e.g. walking robots) is challenging because the design space is highly combinatorial; it is effectively a search over graphs representing kinematic chains. This combinatorial structure has (in prior work) precluded the use of efficient optimization methods over continuous spaces. To solve this challenge, the authors propose to use a graph variational auto-encoder to learn a continuous latent representation of the graph grammar defining a walking robot, then apply Gaussian-process Bayesian optimization to optimize in this continuous latent space. They additionally use a "performance predictor" to regularize the learned latent space to group physically similar robots together. The paper is well-written and includes sufficient experimental evidence to justify the authors' design choices and show an improvement over the state of the art.

**Issues:**

The biggest issue with this paper is that the details of the encoder and decoder architecture are not very clear, particularly to a reader unfamiliar with graph neural networks. It would be helpful to either reference a clear explanation of this message passing process (perhaps a survey or tutorial) or include a more thorough explanation in the supplementary materials.

**Quality Of The Limitations Section:**

Limitations are addressed clearly

**Reviewer Expertise:**

3: The reviewer is fairly confident that the evaluation is correct

**Robotics Focus:**

Relevant but unlikely to deploy to hardware in near future

**Strengths And Weaknesses:**

Strengths:
- The paper is well-written.
- The challenge of optimizing kinematic robots is clear, and the authors' approach is well-motivated and novel.
- The paper contains sufficient experimental evidence to show that the proposed method outperforms the state of the art, and the ablation studies justify the design choices made.

Weaknesses:
- There is little theoretical justification for the effect of performance prediction on the structure of the latent space. I am curious about questions like, "how does the choice of predicted metric impact the structure of the latent space?" and "is the contact configuration the best metric for this task?". The paper provides sufficient empirical results to justify their heuristic, but I am still curious.
- As the authors note in the paper, the walking robots they study are not terribly realistic and might be difficult to construct and deploy in hardware.

**Summary Of Recommendation:**

This paper was fun to read and contains a very interesting core idea (using a learned latent space to enforce a continuous structure on a combinatorial problem) that I hope to see future works build upon.

---

> ### Author Response · Authors · 2022-08-21
> **Response to Reviewer t5Qe**
>
> Thank you for your valuable feedback. We agree that at the current state of our work there is little theoretical reasoning for the choice of predicted metric. In fact, it is possible that the contact point heuristic used in this work is not “optimal” since there could be many different possible heuristics. One important takeaway of this work is that how the design is encoded significantly impacts design optimization.The property predictor gives us a way to inject domain intuition into the design encoding in a manner that ends up improving optimization efficiency. An open question is how to apply a more systematic, automatic, and theoretically derived way of determining the metric(s) used for property prediction.
>
> We agree that extending the work to real world hardware experiments is an important step towards robot design optimization, and would be a key focus of our future work. We added a more thorough discussion about extending our work to real-world hardware in the limitation section of the updated draft.
>
> Regarding the message passing process, we think that the paper of Wu et al. [1] referenced in section 3.2 of our original draft provides a reasonable amount of background knowledge. We also updated section 3.3 of our draft to clarify the decoding process.
>
> [1] Wu, Zonghan, et al. "A comprehensive survey on graph neural networks." IEEE transactions on neural networks and learning systems 32.1 (2020): 4-24.

---

### Official Review · Reviewer_9cM8 · 2022-07-26

**Originality:** Good
**Technical Quality:** Good
**Clarity Of Presentation:** Very Good
**Impact:** 3

**Recommendation:**

Weak Reject: I recommend rejecting the paper, but will not argue for my recommendation if the majority of other reviewers have a different opinion.

**Summary:**

The paper describes an approach to improving the computational efficiency of optimization over discrete, grammar-constrained robot design spaces. Fundamental to this approach is using a variational autoencoder to learn a continuous, low-dimensional representation of the design space that can be optimized using Bayesian optimization. Experimental results show advantages over existing baselines along with the value of training the VAE with an additional objective that encourages the learned representations of similar designs to be nearby in the latent space.

The idea of using a learned latent representation to improve the efficiency of optimization is interesting and offers potential advantages over standard methods that involve combinatorial search. As such, this work may prove useful to others in the robot learning community who are interested in design-control optimization. However, the advantages of this approach in terms of computational efficiency are not fully clear.

**Issues:**

See issues discussed above.

**Quality Of The Limitations Section:**

Limitations are not well addressed

**Reviewer Expertise:**

5: The reviewer is absolutely certain that the evaluation is correct and very familiar with the relevant literature

**Robotics Focus:**

Relevant but unlikely to deploy to hardware in near future

**Strengths And Weaknesses:**

The paper considers the problem of learning the physical design of (legged) robots optimized according to a given (locomotion) task. Building off of recent work, the paper adopts a graph-based representation of the design space that is governed by a grammar. While the grammar has been shown enable heuristic search strategies, the *discrete* design space remains combinatorially large.

The key contribution of this paper is the proposal to learn a low-dimensional, *continuous* design representation that affords efficient optimization using standard optimization methods (i.e., Bayesian optimization). This representation is learned using a graph neural network-based variational autoencoder (VAE) that seeks to reconstruct a collection of input graphs that model designs randomly generated according to the grammar. In addition to the decoder, the network includes a "property predictor" head that takes as input the learned latent representation and outputs the 2D coordinates of a design's ground contact points. The VAE is trained using a loss that combines the standard VAE objective with the MSE of the predicted coordinates. Having learned the latent representation, the authors use Bayesian optimization to reason over candidate designs and use MPC to control each design.

The method is evaluated in simulation on a series of locomotion tasks and compared to several baselines, including the graph heuristic search proposed in the RoboGrammar paper that this work builds on. The paper then ablates the contributions of the VAE prediction objective as well as the grammar.


STRENGTHS

+ The paper considers an important problem that has attracted a fair bit of attention in the robot learning community of-late.
+ The idea of learning a low-dimensional continuous representation of the otherwise discrete design space as a means of improving efficiency is interesting.
+ The paper is well written and easy to follow.

WEAKNESSES

- The core contribution of the paper is the use of a VAE to learn a continuous design representation as a means of improving optimization efficiency. In its current form, however, the paper lacks a concrete demonstration of these gains in efficiency over baseline methods.
- It is not clear what logic was used to choose the maximum steps for training (500). Comparing the results presented here to those in the RoboGrammar paper [5 (in this paper], Guided Heuristic Search (GHS) reaches ~5.5 reward at ~1200 iterations for Frozen Lake, ~6.15 reward at 2000 iterations (and ~6.0 at ~1000 iterations) for Flat Terrain, ~5.25 reward by 2000 iterations for Ridged Terrain, and ~5.6 reward by 2000 iterations (with a rapid increase around 500 iterations). This may make one wonder whether the choice of 500 was cherry picked. What does the reward for GLSO look like farther into training?
- Given the motivation of improving computational efficiency, it would be helpful to see a visualization of performance as a function of computation time (ignoring the fact that methods like the random baseline could be parallelized).
- The means by which contact points are used to define the additional prediction objective is ad hoc and seems brittle (e.g., it relies on a default configuration of each design). It's also not clear how it would generalize to other domains, while the paper claims that the overall method generalizes across domains.
- The paper omits a discussion of related work in design-control optimization [1,2,3]. This includes existing work that uses a VAE to learn a low-dimensional representation that facilitates the optimization of the design and control of soft robots (i.e., by making the full pipeline differentiable) [1]  and another that employs a variational objective to learn a latent representation suited to control and estimation [2]. Also relevant is work that uses graph neural networks to reason over graph-based representations of robot design spaces [3, 4, 5, 6].
- The limitations discussion is weak. The inability to handle acyclic graphs is listed as a limitation without a discussion of the advantages of being able to handle cycles. A more obvious limitation, which the text only partially alludes to, is the challenge of generalizing this approach to produce physically realizable designs that can be fabricated and controlled in the real world.

QUESTIONS/COMMENTS:

* How was the computational cost of the random baseline balanced with the computation afforded the other methods (e.g., since the random baseline can be parallelized)?
* Lines 47–53: I find this discussion a bit contradictory. The paper states that structurally similar robots may have very different performance and uses this as motivation for using contact point prediction as an auxiliary objective. Wouldn't the same structurally similar designs also have similar contact points?
* Can the authors elaborate on the "computationally expensive training procedures" required of end-to-end methods.
* The VAE is trained on a random set of designs generated according to the grammar. Instead, it would be interesting if this optimization could be focused on higher-performing designs.

MINOR:

* Line 29 (and elsewhere): "e.g." should be followed by a comma
* Line 56: Erroneous space before footnote.
* Footnote 1* (and elsewhere): Numbered citations shouldn't be used as words.
* Line 73: "... when the design space are ..." --> "... when the design **spaces** are ..."
* Line 84: "... in a A* manner, ..." --> "... in **an** A* manner, ..."
* Line 153: $\sigma$ is overloaded.
* Line 186: "KL" should be capitalized.
* Line 209: "... uses Gaussian Process" --> "... uses Gaussian **Processes**"


REFERENCES

[1] Andrew Spielberg, Allan Zhao, Tao Du, Yuanming Hu, Daniela Rus, and Wojciech Matusik, "Learning-In-The-Loop Optimization: End-To-End Control And Co-Design of Soft Robots Through Learned Deep Latent Representations," NeurIPS 2019

[2] Andrew Spielberg, Alexander Amini, Lillian Chin, Wojciech Matusik, and Daniela Rus, "Co-learning of task and sensor placement for soft robotics." IEEE Robotics and Automation Letters 6, no. 2 (2021): 1208-1215.

[3] Ye Yuan, Yuda Song, Zhengyi Luo, Wen Sun, and Kris Kitani. "Transform2Act: Learning a Transform-and-Control Policy for Efficient Agent Design," arXiv preprint arXiv:2110.03659 (2021).

[4] Deepak Pathak, Christopher Lu, Trevor Darrell, Phillip Isola, and Alexei A. Efros, "Learning to control self-assembling morphologies: a study of generalization via modularity," NeurIPS 2019.

[5] Tingwu Wang, Renjie Liao, Jimmy Ba, and Sanja Fidler, "Nervenet: Learning structured policy with graph neural networks." ICLR 2018.

[6] Wenlong, Huang Igor Mordatch, and Deepak Pathak, "One policy to control them all: Shared modular policies for agent-agnostic control." ICML 2020.


**Summary Of Recommendation:**

The paper describes an approach to improving the computational efficiency of optimization over discrete, grammar-constrained robot design spaces. Fundamental to this approach is using a variational autoencoder to learn a continuous, low-dimensional representation of the design space that can be optimized using Bayesian optimization. Experimental results show advantages over existing baselines along with the value of training the VAE with an additional objective that encourages the learned representations of similar designs to be nearby in the latent space.

The idea of using a learned latent representation to improve the efficiency of optimization is interesting and offers potential advantages over standard methods that involve combinatorial search. As such, this work may prove useful to others in the robot learning community who are interested in design-control optimization. However, the advantages of this approach in terms of computational efficiency are not fully clear.

UPDATE AFTER AUTHOR RESPONSE/DISCUSSION:

I appreciate the authors' response, which addresses some of my initial questions and concerns.

I remain unconvinced by the justification for the choice to stop at 500 iterations as well as the argument against parallelization. I appreciate that running GLSO for a greater number of iterations and for multiple seeds would take more time, but we aren't talking about a lot of time here, particularly if GLSO was run on a machine more powerful than a. desktop, particularly given that the baseline results already exist. Real-world fabrication is not a focus here---the algorithm is not designed with the specific goal of enabling sim-to-real transfer and there is little discussion about how the challenge of realizing designs that are effective in the real-world would be addressed. This is not a criticism of the work, since it is consistent with the large majority of recent approaches to co-optimization, but calls into question the arguments for stopping at 500 iterations and for not considering parallelization.

As a result, I am keeping my score.

---

> ### Author Response · Authors · 2022-08-21
> **Response to Reviewer 9cM8 (2/2)**
>
> Continuing from (1/2)
>
> > The paper omits a discussion of related work...
>
> We have added the citations correspondingly.
>
> > The limitations discussion is weak. The inability to handle acyclic graphs is listed as a limitation without a discussion of the advantages of being able to handle cycles. A more obvious limitation, which the text only partially alludes to, is the challenge of generalizing this approach to produce physically realizable designs that can be fabricated and controlled in the real world.
>
> Thank you for pointing this out. We removed the limitation about acyclic graphs and added a more thorough discussion about extending our work to real-world hardware in the limitation section of the updated draft.
>
> > QUESTIONS/COMMENTS:
> > How was the computational cost of the random baseline balanced with the computation afforded the other methods (e.g., since the random baseline can be parallelized)?
>
> Indeed, parallelization can decrease the walltime of the random baseline, for evaluation of a candidate design in simulation.
> Notice that Evolutionary algorithms [1] and Bayesian Optimization [2] can also be parallelized. Eventually, we want to be able to perform all evaluations in the real world, where massive parallelization is unlikely. Therefore, in this work we do not consider the effect of parallelization.
>
> [1] Sudholt, Dirk. "Parallel evolutionary algorithms." Springer Handbook of Computational Intelligence. Springer, Berlin, Heidelberg, 2015. 929-959.
>
> [2] Wang, Jialei, et al. "Parallel Bayesian global optimization of expensive functions." Operations Research 68.6 (2020): 1850-1865.
>
> > Lines 47–53: I find this discussion a bit contradictory. The paper states that structurally similar robots may have very different performance and uses this as motivation for using contact point prediction as an auxiliary objective. Wouldn't the same structurally similar designs also have similar contact points?
>
> The similarity of robot structure refers to the similarity of their graph representation. This will be the only information available to the VAE when trained without auxiliary objectives. However, robots with similar graph representation may have very different contact points: consider the case of two hexapods, one with legs pointing upward and the other with legs pointing downward. They will have almost identical graph representation, but their contact point with ground is very different.
>
>
> > Can the authors elaborate on the "computationally expensive training procedures" required of end-to-end methods.
>
> The goal of the end-to-end methods is to train a design generator which maps from a task to a design. Such an objective requires the generator to reason about the relationship between task and designs, and would require massive training samples which would take a long time to collect. We have added a phrase to the related work section to clarify this point.
>
> > The VAE is trained on a random set of designs generated according to the grammar. Instead, it would be interesting if this optimization could be focused on higher-performing designs.
>
> We agree that this is a promising direction which might further improve the performance of GLSO. One open question is where the higher-performing designs would come from, and a potential solution is to utilize an iterative scheme that alternates between design optimization and latent space learning (using designs sampled during the optimization).
>
> > MINOR
>
> We thank the reviewer for pointing out these errors. We have fixed them in the revised draft.

---

> > ### Author Response · Authors · 2022-08-27
> > **Response to Reviewer 9cM8 (additional)**
> >
> > **Comment:**
> >
> > > Given the motivation of improving computational efficiency, it would be helpful to see a visualization of performance as a function of computation time (ignoring the fact that methods like the random baseline could be parallelized).
> >
> > >> Thank you for pointing this out. We are running additional experiments to visualize the performance w.r.t computation time. We will post these additional results once they have completed.
> >
> > Here we provide additional results that demonstrate the wall clock time of different algorithms. We measure compute time for the Flat Terrain and Frozen Lake Terrain tasks, each with three different algorithms: GLSO (ours), GHS (from RoboGrammar), and Random Search. Each algorithm was run three times with different random seeds, and the average is plotted. Please see the attached zip file for results.
> >
> > In FlatTerrain_time.png and FrozenLake_time.png, we present the wall clock time of each algorithm as a function of the optimization steps. We can see that all three lines are roughly linear, and on the same order of magnitude, which we expect given that the evaluation step (MPC in a 3D rigid body dynamics simulation) is the major computational bottleneck of the optimization process.
> > In FlatTerrain_reward.png and FrozenLake_reward.png, we plot rewards with respect to the optimization time. Each algorithm was run for 500 steps, so each method on the plot has a different ending time. For both tasks, random search completes 500 iterations fastest, and GHS slowest. Compared to random search, the additional computation time of GHS comes from training a GNN evaluation model, as well as performing search based on the trained evaluation model. The additional computation time of GLSO comes from fitting a Gaussian Process model to the current points, as well as evaluating the acquisition function.
> >
> > This experiment further supports our conclusion that GLSO is a more computationally efficient method to search for task-specific robot designs than GHS and random search.
> >
> >
> > **Zip File:**
> >
> > /attachment/7b27f4b8ce8372115e1ff4b6b01dfa6863628da1.zip

---

> ### Author Response · Authors · 2022-08-21
> **Response to Reviewer 9cM8 (1/2)**
>
> Thank you for your valuable feedback! We respond to your main concerns individually below:
>
> > WEAKNESSES
>
> > The core contribution of the paper is the use of a VAE to learn a continuous design representation as a means of improving optimization efficiency. In its current form, however, the paper lacks a concrete demonstration of these gains in efficiency over baseline methods.
>
> > It is not clear what logic was used to choose the maximum steps for training (500). Comparing the results presented here to those in the RoboGrammar paper [5 (in this paper], Guided Heuristic Search (GHS) reaches ~5.5 reward at ~1200 iterations for Frozen Lake, ~6.15 reward at 2000 iterations (and ~6.0 at ~1000 iterations) for Flat Terrain, ~5.25 reward by 2000 iterations for Ridged Terrain, and ~5.6 reward by 2000 iterations (with a rapid increase around 500 iterations). This may make one wonder whether the choice of 500 was cherry picked. What does the reward for GLSO look like farther into training?
>
> For the two concerns above: The goal of this work is to identify task-specific designs with a limited number of samples. When the evaluation is conducted in simulation, we found that running for 2000 iterations takes 3 - 4 days on a regular desktop computer.  We chose the maximum steps of optimization as 500 because it corresponded to simulation evaluations which take a full day of computation on a desktop, and each experiment was run for multiple trials and for multiple tasks. We hope that future work can build towards design optimization with even fewer samples, which we believe will be an important step towards eventually deploying it in the real world.
> Nevertheless, we agree that showing the asymptotic outputs, where each method is allowed nearly unlimited computation, may provide the reader with a better understanding of the behavior of GLSO and related approaches.
>
>
> > Given the motivation of improving computational efficiency, it would be helpful to see a visualization of performance as a function of computation time (ignoring the fact that methods like the random baseline could be parallelized).
>
> Thank you for pointing this out. We are running additional experiments to visualize the performance w.r.t computation time. We will post these additional results once they have completed.
>
> > The means by which contact points are used to define the additional prediction objective is ad hoc and seems brittle (e.g., it relies on a default configuration of each design). It's also not clear how it would generalize to other domains, while the paper claims that the overall method generalizes across domains.
>
> We agree that at the current state of our work there is little theoretical reasoning for the choice of predicted metric. In fact, it is possible that the contact point heuristic used in this work is not “optimal” since there could be many different possible heuristics. One important takeaway of this work is that how the design is encoded significantly impacts design optimization.The property predictor gives us a way to inject domain intuition into the design encoding in a manner that ends up improving optimization efficiency. An open question is how to apply a more systematic, automatic, and theoretically derived way of determining the metric(s) used for property prediction.

---

### Official Review · Reviewer_tfFP · 2022-07-27

**Originality:** Very Good
**Technical Quality:** Good
**Clarity Of Presentation:** Very Good
**Impact:** 4

**Recommendation:**

Weak Accept: I recommend accepting the paper, but will not argue for my recommendation if the majority of other reviewers have a different opinion.

**Summary:**

This paper presents a robot co-design algorithm that co-optimizes the shape and control of terrestrial robots. This paper is based upon the RoboGrammar paper [5] and uses grammar to constrain the robot topology search space to be compact. The key idea of this paper is to train an autoencoder and decoder for the robot structure, which maps the discrete robot topology into a continuous latent space representation. This encoder-decoder framework enables to optimize in the continuous latent space with bayesian optimization instead of optimizing in the discrete and constrained robot topology space. The authors demonstrate the advantages of the proposed method for the tasks defined in [5] and properly design the ablation study to validate each technical component of the method.

**Issues:**

All the issues have been mentioned in the above.

**Quality Of The Limitations Section:**

Limitations are addressed clearly

**Reviewer Expertise:**

5: The reviewer is absolutely certain that the evaluation is correct and very familiar with the relevant literature

**Robotics Focus:**

Relevant but unlikely to deploy to hardware in near future

**Strengths And Weaknesses:**

The idea of mapping a discrete topology space into a continuous latent space to simplify the robot design optimization problem is interesting. The method to decode the continuous latent space into the robot structure with a depth-first approach and using teacher-force style training is inspiring. The most appealing part of the algorithm to me is the reusability of the trained autoencoder, which has the potential to apply the pre-trained autoencoder to other tasks/problems.

However, there are a few concerns and questions I have for the authors:
1. To train the autoencoder, the authors collect a large dataset of designs by applying random rules in the RoboGrammar. However, randomly picking the rules to apply will result in a super unbalanced dataset, where the number of simple designs with short rule sequences will dominate the whole dataset. The autoencoder trained with such an unbalanced dataset will probably lead to a biased encoder that only generates simple designs. Does the author do any data balance technique to address this problem?

2. The robot decoder part (section 3.3) is the most important part of the whole pipeline since it requires a clever way to convert a continuous latent space to valid discrete structure space. More details about the decoder will be necessary for a better understanding and reproducibility of the approach. Specifically, I have the following questions regarding the decoder part:
     1. How does the recorder message passing work? Since the partial graph keeps changing during the decoding phase, do the messages stored at each node get cleared and propagate from scratch every round?

     2. One of the advantages of using grammar is to constrain the robot designs to be valid and manufacturable designs. However from the paper, during the decoding phase, it seems that the algorithm just iteratively adds a node with a certain type to the current partial graph without any verification on whether the design is actually spanned by the grammar. If that is the case, the produced designs can be invalid for real manufacturing thus the applicability of the proposed method to a real robot system is questionable.

     3. In RoboGrammar, the grammar rules are constructed to provide the symmetry of the robot structure. The designs in the associated video also have such symmetry property. From the paper, I didn’t find such symmetry constraints in the decoder part. Could you clarify more on this part?

     4. While it is appealing to convert a discrete parameter space (i.e. robot topology space) into a continuous space (i.e. latent space), it is counter-intuitive. While there are gradients in the continuous space, the gradients are not defined in the discrete space. As an example, if we slightly perturb the latent representation, how will the robot structure produced by the decoder change?

     5. It is good to visualize the learned latent space. You can take two latent vectors and get intermediate latent vectors by interpolating between these two vectors, and visualize what the corresponding robot structures after decoding look like.

     6. How is the ground truth label is constructed? For a node, if it has multiple other nodes connected to it (e.g. the root node), then there can be multiple choices for the label of the first node starting from this node, which means the ground truth label q is not unique.

     7. The training takes place in a teacher-force style, it will produce aggregation errors during testing. How do the authors resolve this problem?

3. The wall terrain task looks different from the one in RoboGrammar [5] (the wall is narrower in this work), any reasons for that?

4. The comparison to GHS looks questionable. In [5], GHS can achieve much higher rewards after training for 2000 iterations (e.g. over 6 in Flat Terrain Task), which means both the GHS and GLSO have not reached the best design yet after 500 iterations. It would be helpful to train the GLSO and GHS for a long enough time to compare their convergence speed and final performance.

5. How long does the algorithm run? The major advantage of training an autoencoder is that it can generalize to different new tasks without re-training the encoder and saves time. So it would be more supportive if the robot optimization time can be reported.

6. Missing references:
    1. Evolutionary Algorithms:
        > Embodied intelligence via learning and evolution, Gupta et al. 2021

        > Evolution Gym: A Large-Scale Benchmark for Evolving Soft Robots, Bhatia et al. 2021

        > Evolving virtual creatures, Sims 1994

    2. RoboGrammar: citation [10] should be included in section 2.2.

    3. Latent Space Optimization:
        > Emergent hand morphology and control from optimizing robust grasps of diverse objects, Pan et al. 2020




**Summary Of Recommendation:**

Given the novelty of the method and the potentially promising results and applications, I would suggest weak acceptance for now and will change my score accordingly based on whether the authors address my concerns properly.

---

> ### Author Response · Authors · 2022-08-21
> **Response to Reviewer tfFP (2/2)**
>
> **Comment:**
>
> Continuing from (1/2)
>
> >While it is appealing to convert a discrete parameter space (i.e. robot topology space) into a continuous space (i.e. latent space), it is counter-intuitive. While there are gradients in the continuous space, the gradients are not defined in the discrete space. As an example, if we slightly perturb the latent representation, how will the robot structure produced by the decoder change?
>
> We do not use gradients during optimization because Bayesian Optimization does not require them, and we don’t have access to the gradient of the objective function (in this case, the reward function based on the distance traveled by each design). Converting the space from discrete to continuous opens up the possibility of using continuous optimization methods like BO without being limited to algorithms for discrete decision spaces. Some examples of perturbation in the latent space can be seen in Figure 3 of our paper, and some additional latent space interpolation visualizations are provided in the zip file attached.
>
>
> >It is good to visualize the learned latent space. You can take two latent vectors and get intermediate latent vectors by interpolating between these two vectors, and visualize what the corresponding robot structures after decoding look like.
>
> Thank you for the suggestion. To aid in the intuition behind the latent space, we have created 4 sets of visualization of linear interpolation in the latent space, shown by interpolation.png in the attached images. The left and right-most designs in each row are obtained from two points in latent space, and the designs between them are created by linearly interpolating the latent vector at equal intervals, then decoding those variables. These images have also been included in the updated supplementary material.
>
> >How is the ground truth label is constructed? For a node, if it has multiple other nodes connected to it (e.g. the root node), then there can be multiple choices for the label of the first node starting from this node, which means the ground truth label q is not unique.
>
> Indeed, the order between sibling nodes is ambiguous. In this work, the order is determined through a depth-first traversal of the input design graph, which gives us the ground truth label for each node. We agree that this process can be improved in future work through applying permutation-invariant techniques during the generation of graphs.
>
>
> >The training takes place in a teacher-force style, it will produce aggregation errors during testing. How do the authors resolve this problem?
>
> We did not apply specific methods to address aggregation errors, as we did not observe them to occur with the decoder during testing.
>
> >The wall terrain task looks different from the one in RoboGrammar [5] (the wall is narrower in this work), any reasons for that?
>
> RoboGrammar has different versions of tasks (i.e. WallTerrainTask v.s NewWallTerrainTask). The NewWallTerrainTask has wider walls compared to the WallTerrainTask. We use the WallTerrainTask version, building on the RoboGrammar code repository.
>
> >The comparison to GHS looks questionable. In [5], GHS can achieve much higher rewards after training for 2000 iterations (e.g. over 6 in Flat Terrain Task), which means both the GHS and GLSO have not reached the best design yet after 500 iterations. It would be helpful to train the GLSO and GHS for a long enough time to compare their convergence speed and final performance.
>
> The goal of this work is to identify task-specific designs with a limited number of samples. When the evaluation is conducted in simulation, we found that running for 2000 iterations takes 3 - 4 days on a regular desktop computer.  We chose the maximum steps of optimization as 500 because it corresponded to simulation evaluations which take a full day of computation on a desktop, and each experiment was run for multiple trials and for multiple tasks. We hope that future work can build towards design optimization with even fewer samples, which we believe will be an important step towards eventually deploying it in the real world.
> Nevertheless, we agree that showing the asymptotic outputs, where each method is allowed nearly unlimited computation, may provide the reader with a better understanding of the behavior of GLSO and related approaches.
>
>
> >How long does the algorithm run? The major advantage of training an autoencoder is that it can generalize to different new tasks without re-training the encoder and saves time. So it would be more supportive if the robot optimization time can be reported.
>
> We are running additional experiments to visualize the performance w.r.t computation time. We will post these additional results once they have completed.
>
> >Missing references
>
> We thank the reviewer for pointing out these related work, and have added the citations correspondingly.
>
> **Zip File:**
>
> /attachment/307113218fa92e8f37f8239288403ebb8da7a774.zip

---

> > ### Author Response · Authors · 2022-08-27
> > **Response to Reviewer tfFP (additional)**
> >
> > **Comment:**
> >
> > > How long does the algorithm run? The major advantage of training an autoencoder is that it can generalize to different new tasks without re-training the encoder and saves time. So it would be more supportive if the robot optimization time can be reported.
> >
> > >>We are running additional experiments to visualize the performance w.r.t computation time. We will post these additional results once they have completed.
> >
> > Here we provide additional results that demonstrate the wall clock time of different algorithms. We measure compute time for the Flat Terrain and Frozen Lake Terrain tasks, each with three different algorithms: GLSO (ours), GHS (from RoboGrammar), and Random Search. Each algorithm was run three times with different random seeds, and the average is plotted. Please see the attached zip file for results.
> >
> > In FlatTerrain_time.png and FrozenLake_time.png, we present the wall clock time of each algorithm as a function of the optimization steps. We can see that all three lines are roughly linear, and on the same order of magnitude, which we expect given that the evaluation step (MPC in a 3D rigid body dynamics simulation) is the major computational bottleneck of the optimization process.
> > In FlatTerrain_reward.png and FrozenLake_reward.png, we plot rewards with respect to the optimization time. Each algorithm was run for 500 steps, so each method on the plot has a different ending time. For both tasks, random search completes 500 iterations fastest, and GHS slowest. Compared to random search, the additional computation time of GHS comes from training a GNN evaluation model, as well as performing search based on the trained evaluation model. The additional computation time of GLSO comes from fitting a Gaussian Process model to the current points, as well as evaluating the acquisition function.
> >
> > This experiment further supports our conclusion that GLSO is a more computationally efficient method to search for task-specific robot designs than GHS and random search.
> >
> >
> > **Zip File:**
> >
> > /attachment/3c233ee8f2412bcc5795ca8b3f836f146eff662a.zip

---

> > > ### Comment · Reviewer_tfFP · 2022-08-28
> > > **Response to the comments**
> > >
> > > Thanks the authors for the detailed clarification and the additional experiments/plots. The interpolation plot really helps to understand the quality of the latent space, so I think including it in the paper would be nice. The authors respond to most of my clarity problems  (i.e. the unbalance problem of the randomly generated dataset, the mechanism of the decoder), and it is surprising to me that the results are always reasonable even if the authors do not put too much effort into constraining the system. As for running the algorithm for only 500 iterations, I am not sure whether 3-4 days for a robot design optimization is a long time since a typical manual robot design process will take several months to a year on optimizing the robot. In that sense, spending several days on getting a good-performing robot should be reasonable for practical robot design problems in my mind.  Thus I encourage the authors to include the experiment of running 2000 iterations in the final version to see whether it can actually converge to the designs as good as baselines to give the audience a good sense of the pros and (probable) cons of the proposed approach. Though it is not perfect yet, I believe this paper can contribute good value to the robot co-design field, so I would like to keep my original assessment.

---

> ### Author Response · Authors · 2022-08-21
> **Response to Reviewer tfFP (1/2)**
>
> **Comment:**
>
> Thank you for your valuable feedback! We respond to your main concerns individually below:
>
> >To train the autoencoder, the authors collect a large dataset of designs by applying random rules in the RoboGrammar. However, randomly picking the rules to apply will result in a super unbalanced dataset, where the number of simple designs with short rule sequences will dominate the whole dataset. The autoencoder trained with such an unbalanced dataset will probably lead to a biased encoder that only generates simple designs. Does the author do any data balance technique to address this problem?
>
> We thank the reviewer for this insightful observation. Indeed, when developing our method we found that randomly generating designs using grammar rules without any post-processing results in a dataset with simple designs over-represented, as shown by distribution.png which we created and attached in additional_imgs.zip.
>
> Empirically, we found that pruning out designs with less than 6 modules was sufficient to learn a meaningful latent space, without applying other data balancing techniques. If future adaptations of this work find that this simple pruning technique does not sufficiently re-balance the dataset, one could explore whether techniques based on weighted sampling could help with the creation of a “better” latent space, which may impact downstream optimization.
>
> >The robot decoder part (section 3.3) is the most important part of the whole pipeline since it requires a clever way to convert a continuous latent space to valid discrete structure space. More details about the decoder will be necessary for a better understanding and reproducibility of the approach. Specifically, I have the following questions regarding the decoder part:
>
> >How does the recorder message passing work? Since the partial graph keeps changing during the decoding phase, do the messages stored at each node get cleared and propagate from scratch every round?
>
> The messages of existing nodes will not be cleared during decoding. When each new node is created, its message is initialized based on the message of its parent. We added a sentence in the updated draft (line 168) to make this point clear.
>
> >One of the advantages of using grammar is to constrain the robot designs to be valid and manufacturable designs. However from the paper, during the decoding phase, it seems that the algorithm just iteratively adds a node with a certain type to the current partial graph without any verification on whether the design is actually spanned by the grammar. If that is the case, the produced designs can be invalid for real manufacturing thus the applicability of the proposed method to a real robot system is questionable.
>
> >In RoboGrammar, the grammar rules are constructed to provide the symmetry of the robot structure. The designs in the associated video also have such symmetry property. From the paper, I didn’t find such symmetry constraints in the decoder part. Could you clarify more on this part?
>
> For the two concerns above: The constraint for symmetry and validity (as defined by the grammar rules) is implicit with respect to the decoder - there is no explicit constraint that forbids the decoder from generating invalid/asymmetric designs. However, the dataset only contains symmetric valid designs, and the decoder learns to only generate designs similar to the training data. Empirically, we find that the decoder quickly learns to only generate symmetric and plausible designs during training.
> Explicitly constraining the output space of the decoder could ensure that only valid designs can be generated, and may expedite the training of the VAE. This would be a potential variation to our methods if future adaptations of this work encounter frequent invalid designs output by the decoder.
>
>
>
>
> **Zip File:**
>
> /attachment/f992b12dca5cf2b5f66e08c05483712ebded3021.zip

---

### Official Review · Reviewer_4uZ5 · 2022-08-01

**Originality:** Fair
**Technical Quality:** Very Good
**Clarity Of Presentation:** Very Good
**Impact:** 3

**Recommendation:**

Weak Accept: I recommend accepting the paper, but will not argue for my recommendation if the majority of other reviewers have a different opinion.

**Summary:**

The paper describes a method for optimizing a robot design given a task.

The robot topology is defined as a graph which, in turn, is generated according to a grammar that guarantees only valid robot topologies.

The grammar is also used to generate a synthetic Dataset of 500k robot topologies, used to train a VAE mapping the graph structure of the robots to a continuous latent space. An additional constraint is imposed that similar points in the latent space produce robots with a similar “shape”.

Bayesian Optimization is used to search over the latent space to generate the robot with the highest performance.
Generated robots are evaluated in Sim against existing baselines.

The controller is computed using a variation of MPC which is the most computationally expensive part of the algorithm.

An ablation study is performed to show the effectiveness of the auxiliary parameter (the Property Predictor) in conjunction with the VAE to locate similar design in the latent space.

**Issues:**

The paper lacks an “on robot” experiment; albeit not strictly required, a validation on simple robots made with cheap servo motors and 3D printed limbs could provide stronger evidence on the quality of the algorithm.

Other than the above mentioned issue the Reviewer could not find any meaningful shortcoming to the proposed work.


**Quality Of The Limitations Section:**

Limitations are addressed clearly

**Reviewer Expertise:**

3: The reviewer is fairly confident that the evaluation is correct

**Robotics Focus:**

Highly relevant to robotics but no hardware experiments

**Strengths And Weaknesses:**

## Strengths:

* The paper is very well written, It’s clear, builds on solid foundations yet performs quite well over other benchmarks.

* The idea of training a continuous latent space on a variety of robots and being able to use it to optimize for other tasks or other domains altogether (manipulation and so forth) seems particularly powerful and promising.

## Weakness:

* No hardware experimentation

**Summary Of Recommendation:**

The paper is very well written, It’s clear, builds on solid foundations yet provides a meaningful increase over the performance metric. The Authors supply the Code to run the experiments and allow the community to keep building upon this work.

---

> ### Author Response · Authors · 2022-08-21
> **Response to Reviewer 4uZ5**
>
> Thank you for your valuable feedback. We agree that extending the work to hardware experiments is an important step towards confirming the applicability of robot design optimization to reality. We added additional discussion about extending our work to real-world hardware to the limitation section of the updated draft, attached as pdf.

---

### Comment · Area_Chair_x7qP · 2022-08-19
**Meta Review Comments**

Thank you authors and reviewers, here is a short summary of some of the key strengths and weaknesses identified by the reviewers.
Authors & reviewers, please engage in a discussion regarding the issues raised by the individual reviews.

Strengths:
- the papers was considered well presented
- the papers overall approach, incorporating a low dimensional continuous latent space for this problem, was considered promising

Weaknesses:
- related work in design-control optimization should be discussed (see reviewer 9cM8) and limitations should be highlighted
- the lack of hardware experimentation/validation as well as potential difficulties in realizing the simulated robot designs in reality
- it was questioned if the application of random rules in the RoboGrammar may be too simplistic, resulting in a dataset with few complex designs to train on
- additional details regarding the decoder in section 3.3 were requested as well as a clearer statement of computational cost and number of training iterations and how these were chosen.

---

> ### Author Response · Authors · 2022-08-28
> **Response to Area Chair x7qP**
>
> Thank you so much for the summary. We have replied to the reviewers in their corresponding threads.

---

### Meta-Review · Area_Chair_x7qP · 2022-09-05

**Recommendation:** Accept (Poster)
**Confidence:** 3

**Metareview:**

Concluding comments and observations:
The authors present their work clearly and report improvements over baseline evolutionary design approaches.

Pros:
- The work tackles a very challenging robot design optimization problem featuring both discrete and continuous optimization aspects which are tackled in an interesting manner in the proposed architecture with promising numerical results.
-  I personally think the work has merit also as a promising direction for future work to build upon and may generate discussion among the CoRL community.

Cons:
- No real world evaluation of robot designs. All results based on simulation alone.
- Success/Quality of a robot design is considered in terms of success of learning MPC controller for certain tasks only.
- the question of weather evaluation up to only 500 designs/iteration steps is sufficient requires further investigation

The reviewers' opinion was somewhat divided between between 1x strong accept, 2x weak accept and 1x weak accept. While there are certainly aspects that could be further improved, I lean slightly towards accept given the potential of the proposed methodology and its future improvements to robot design.